# Diverse patterns of correspondence between protist metabarcodes and protist metagenome-assembled genomes

**Daryna Zavadska[1]***, **Nicolas Henry[2]**, **Adrià Auladell[1]**, **Cédric Berney** [3], **Daniel J. Richter[1]***

**1** Institut de Biologia Evolutiva (CSIC-Universitat Pompeu Fabra), Barcelona, Spain, **2** CNRS, FR2424, ABiMS, Station Biologique de Roscoff, Sorbonne Université, Roscoff, France, **3** CNRS, UMR7144, AD2M, Station Biologique de Roscoff, Sorbonne Université, Roscoff, France

* zavadskadaryna@gmail.com (DZ); daniel.j.richter@gmail.com (DJR)

**Data Availability Statement:** The data on SMAG abundance and SMAG taxonomic assignment was

## Abstract

Two common approaches to study the composition of environmental protist communities are metabarcoding and metagenomics. Raw metabarcoding data are usually processed into Operational Taxonomic Units (OTUs) or amplicon sequence variants (ASVs) through clustering or denoising approaches, respectively. Analogous approaches are used to assemble metagenomic reads into metagenome-assembled genomes (MAGs). Understanding the correspondence between the data produced by these two approaches can help to integrate information between the datasets and to explain how metabarcoding OTUs and MAGs are related with the underlying biological entities they are hypothesised to represent. MAGs do not contain the commonly used barcoding loci, therefore sequence homology approaches cannot be used to match OTUs and MAGs. We made an attempt to match V9 metabarcoding OTUs from the 18S rRNA gene (V9 OTUs) and MAGs from the *Tara* Oceans expedition based on the correspondence of their relative abundances across the same set of samples. We evaluated several metrics for detecting correspondence between features in these two datasets and developed controls to filter artefacts of data structure and processing. After selecting the best-performing metrics, ranking the V9 OTU/MAG matches by their proportionality/correlation coefficients and applying a set of selection criteria, we identified candidate matches between V9 OTUs and MAGs. In some cases, V9 OTUs and MAGs could be matched with a one-to-one correspondence, implying that they likely represent the same underlying biological entity. More generally, matches we observed could be classified into 4 scenarios: one V9 OTU matches many MAGs; many V9 OTUs match many MAGs; many V9 OTUs match one MAG; one V9 OTU matches one MAG. Notably, we found some instances in which different OTU-MAG matches from the same taxonomic group were not classified in the same scenario, with all four scenarios possible even within the same taxonomic group, illustrating that factors beyond taxonomic lineage influence the relationship between OTUs and MAGs. Overall, each scenario produces a different interpretation of V9 OTUs, MAGs and how they compare in terms of the genomic and ecological diversity they represent.

taken from: https://doi.org/10.1016/j.xgen.2022.100123 Supplementary Data. SMAG sequences were obtained from: https://www.genoscope.cns.fr/tara/V9 raw abundance dataset for V9 metabarcodes organised at OTU level was obtained from: https://doi.org/10.5281/zenodo.7236051 All the intermediate and accessory datasets generated and used in our manuscript, along with the code and plots are available at https://github.com/beaplab/Protist_barcode_MAG_correspondence.

**Funding:** This project has received funding from the European Research Council (ERC) under the European Union's Horizon 2020 research and innovation programme (grant agreement No. 949745). We also acknowledge support from the Departament de Recerca i Universitats de la Generalitat de Catalunya (exp. 2021 SGR 00751).

**Competing interests:** The authors have declared that no competing interests exist.

## Introduction

Shotgun metagenomics and amplicon sequencing (also known as metabarcoding) are commonly used approaches to characterise the composition of environmental microbial communities [1]. The vast majority of protists (microbial eukaryotes) remains uncultured, with cultured representatives largely concentrated in a small number of lineages of the eukaryotic tree of life [2]. Community sequencing approaches are culture-free [3], making them the best currently available method for functional and taxonomic studies of complex communities, as well as selected organisms of interest that cannot be sustained or replicated in the lab [1, 3].

Shotgun metagenomics is the non-targeted sequencing of the DNA present in an environmental sample [4–6]. Metagenome sequencing reads can be processed to obtain Metagenome-Assembled Genomes (MAGs) [7], permitting the reconstruction of metabolic capabilities, population genetics and other characteristics based on individual MAGs. To obtain these MAGs, contigs are assembled *de novo* from whole-community sequence reads and subsequently separated into bins representing individual genomes [3].

Metabarcoding relies on sequencing a specific marker gene or region, which is selectively amplified from environmental samples prior to sequencing [6]. For eukaryotes, the 18S ribosomal RNA gene, and, in particular, its V4 and V9 regions, are the most commonly used marker sequences in metabarcoding [8–12], due to the fact that they are evolutionarily labile regions bounded by regions of near-universal conservation that can be used as target sequences for primers for PCR amplification. Here, we will analyse V9 metabarcoding data, in which PCR-amplified V9 sequence reads have been clustered into V9 Operational Taxonomic Units (OTUs; groups of identical or nearly-identical sequences) via Swarm [13] followed by distribution-based post-clustering curation [14].

These approaches present differing strengths and weaknesses. MAGs are informative in terms of the functional gene repertoire of the community [15], and can also be used for biodiversity assessment and taxonomic characterization [7, 16, 17], but the depth of these assessments can vary greatly, as the coverage is frequently insufficient to detect the taxa with low relative abundance [18]. Metabarcoding, in comparison, offers comparatively precise information in terms of taxonomic composition [7, 17, 19], but it has a relatively limited capacity to predict the detailed functional diversity in the community, because the prediction depends on the availability of closely related reference sequences, and similarity in marker sequences does not guarantee similarity in genomic features [3].

Both individual MAGs and individual V9 OTUs represent groupings of underlying biological entities that share the same sequence. In the case of MAGs, this corresponds to the assembled genome, and for V9 OTUs, a set of identical (or nearly-identical) V9 sequences. Although the taxonomic resolution of MAGs and V9 OTUs may differ greatly, both are generally assumed to represent proxies for groupings with an interpretable biological meaning: for example, species, ecotypes or populations [1, 15, 20–22]. Here, we will use the term "biological entity" as a shorthand to describe the underlying biological reality that either MAGs or V9 OTUs are generally assumed to represent.

Although there are some well-known cases in which either MAGs [23, 24] or V9 OTUs [25–33] do not correspond to a single biological entity, and some broad-scale studies showing that for high-rank taxa MAGs or V9 OTUs generally match a given set of cell morphologies [11, 34, 35], we still lack a global overview of how **individual** MAGs or V9 OTUs correspond to one another and to the **individual** groups of biological entities that are present in a sampled community.

Investigating the relationship between MAGs and V9 OTUs is of further interest as there is an ongoing debate regarding which of the two approaches is more accurate in representing

underlying biological entities [7, 36–38]. Previous comparisons of metagenomics and metabarcoding data [23, 39, 40] have focused on selected groups for which high-quality reference sequences are available, whereas others compared only the diversity of higher-rank taxa, with no attempt to match individual MAGs to metabarcoding OTUs [41], or focused on prokaryotes [42].

There is no direct way to test if MAGs or V9 OTUs represent a single biological entity in the absence of a pre-existing reference genome that contains both sequences, since 18S rRNA sequences (and hence, the 18S V9) are expected to be absent from MAG assemblies [15]. The three principal criteria based on which contigs from metagenomes are binned into MAGs are (1) the differential coverage of reads, (2) average k-mer frequency [4, 7, 43] and (3) GC content [7]. The 18S rRNA gene is a very highly conserved sequence, therefore, it is not possible to bin it into a MAG by discriminating among sequences based on the second and third criteria of k-mer frequency or GC content. Moreover, the 18S is often present in multiple copies inside a genome, especially in protists (e.g., [9, 11]), so, without prior knowledge of copy number, the 18S sequence also cannot be binned into MAGs based on the first criterion of differential coverage information.

Therefore, it is not possible to search for V9 OTU sequences contained in MAG assemblies in order to relate MAGs with V9 OTUs. As an alternative, if some MAGs and some V9 OTUs represent the same underlying biological entity, the relative abundances of MAGs and V9 OTUs should display the same patterns of variation across the samples for which both MAG and V9 OTU datasets were generated. Based on this correspondence, we can match MAG and V9 OTUs in pairs and hypothesise the underlying biological entity they represent.

In this study, we develop a methodology to search for MAGs and V9 OTUs that match to one another, and therefore likely represent the same biological entity, by measuring correspondence between MAG and V9 OTU relative abundance counts across multiple samples Fig 1.

We found that, in addition to the cases of a single MAG matching a single V9 OTU, other scenarios exist: a single MAG can match with more than one V9 OTU and vice versa, and a set of MAGs can match with a set of V9 OTUs Fig 1. Each of these scenarios leads to different hypotheses regarding the nature of the underlying biological entities that they represent.

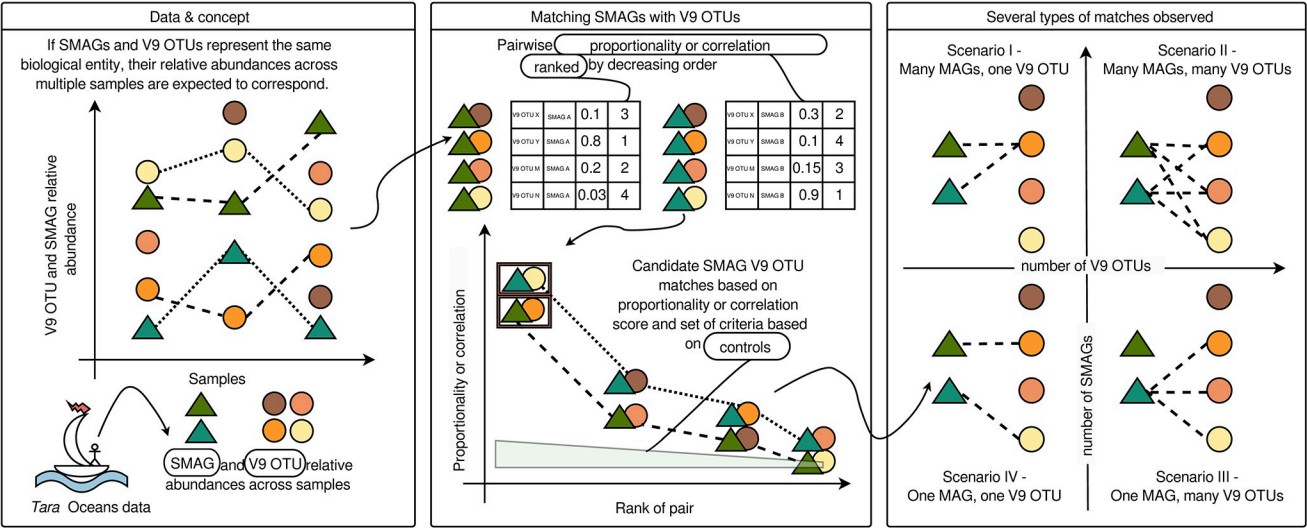

**Fig 1. A graphical outline of the hypothesis, methods and results of this study.**

## Materials and methods

### Development of a method to match MAGs to OTUs based on known positive controls, simulated data, and shuffling

For our analysis, we used the V9 metabarcoding dataset from [14] and MAG and SAGs dataset from [15].

The V9 metabarcoding dataset we analysed was produced by clustering PCR-amplified V9 sequence reads into V9 Operational Taxonomic Units (OTUs; groups of identical or nearly-identical sequences) via Swarm [13] followed by distribution-based post clustering curation [14].

We analysed abundance data from metagenome sequence reads mapped to both MAGs and to Single-cell Amplified Genomes (SAGs). Following Delmont et al., 2022, we will use the term SMAGs to refer to both MAGs and SAGs together. The SAGs, which may contain 18S sequences, were used as a positive control.

Together, the datasets we used contain V9 OTU [14, 48] and SMAG [15] abundances across 705 separate samples for which both the V9 region and metagenomes were sequenced.

For our analyses, we calculated relative abundances of SMAGs and V9 OTUs from the abundances provided in both input datasets. In addition to abundances, both input datasets provided taxonomic assignments. We used these assignments to iteratively partition both datasets into "taxonomic subsets": sets containing both SMAGs and V9 OTUs that were classified into the same taxonomic group at a given rank (subsets with higher ranks include the same number of SMAGs but more V9 OTUs; see S1 Text, S1 Fig, S1 Table in S1 File for more detail). For each subset, we calculated correlation and proportionality from the relative abundances of all possible SMAG-V9 OTU pairs within the subset. Based on these values, for each SMAG within the subset, we ranked the V9 OTUs from the first-best match ("top-1") with the highest correlation or proportionality value to the last Fig 1, S1 Fig, S1 Text in S1 File.

Although the correlation coefficient (in our case, non-parametric Spearman's correlation) is a commonly used measure of the correspondence between the two features in two datasets, alternative methods have been proposed to be more appropriate for the analysis of compositional data (such as V9 OTU and MAG abundance). Proportionality differs from Spearman's rank correlation in that, instead of operating on mean counts, (i) it is calculated based on the variation of relative counts, (ii) it is optimised for log-transformed values and (iii) it is sub-compositionally coherent (i.e., its results should not change in any given subset of the data when compared to the complete dataset) [44–46]. Several variations of the proportionality coefficient, called *rho* ($\rho$), *phi* ($\varphi$), *phs* ($\varphi_s$) and *vlr* have been proposed [45, 47].

Using the methodology described above, we began by testing the performance of Spearman's rank correlation and 4 measures of proportionality: *rho*, *phi*, *phs* and *vlr* [45, 47]. We performed our tests by identifying positive controls in the SMAGs dataset: either SAGs that contained V9 sequences in their assemblies or MAGs whose sequence was identical or nearly identical to a publicly available reference genome containing one or more V9 sequences. In particular, we checked whether the V9 sequence present in the SAG or matching reference genome assembly was the same as the V9 OTU identified as a high-rank match to the SMAG S4 Text in S1 File. Metrics other than *rho* proportionality or Spearman's correlation performed poorly, failing to recover V9 OTUs that matched the V9 sequence found in the genome assembly S2 Fig in S1 File.

We performed a second control by simulating compositional datasets with parameters chosen to be similar to the original SMAG and V9 OTU abundance datasets, in terms of dataset dimension, proportion of 0s in each sample, and variance of abundance counts of features. In each simulated dataset, a single pair of features (i.e., a single SMAG-V9 OTU pair) was artificially created to be proportional/correlated (positive control), and the rest of the features were

noise (negative control) S2 Text in S1 File. All metrics other than Spearman's correlation and *rho* proportionality failed to recover the positive control pair of features as the best-ranked match S6 Fig in S1 File.

Based on the reference genome and simulation controls, we concluded that *rho* proportionality or Spearman's correlation are the most reliable metrics to estimate correspondence between the SMAG and V9 OTU datasets. The results shown in Fig 2 are the ones obtained from *rho* proportionality metrics, with values from Spearman's correlation shown in S2B, S3B, S4, S5B and S10 Figs in S1 File. The selection of candidate MAG-V9 OTU matches described below was based on both *rho* proportionality and Spearman's correlation; the agreement between the results from these two metrics was a criterion for the selection of candidate matches.

Continuing with SAG reference and simulation controls, we next examined whether the correlation or proportionality values within the set of V9 OTU matches to a given SMAG could be used to differentiate among the matches. As described above, we ranked the V9 OTU matches for each SMAG in descending order by pairwise correlation or proportionality value. Based on the difference in the correlation or *rho* proportionality values observed between known true SMAG-V9 OTU matches and known false matches of lower ranks from both SAG-reference genomes and simulations, we established that the minimum difference in correspondence values between pairs of adjacent rank should be > 0.05 for the better-ranked pair to be considered as the more probable candidate match when compared with the pair of worse rank. These differences can be seen in proportionality plots as the slope of the line connecting two dots across the x-axis S1 Fig in S1 File: the larger the slope, the bigger the difference in proportionality. We use the word "breakpoint" to refer to differences of > 0.05 in correlation or proportionality estimates between pairs of adjacent rank S1 Text in S1 File.

We also implemented a third type of control, in which we randomly shuffled the V9 OTU relative abundance dataset in order to obtain baseline "noise" correlation and proportionality values that would be produced by chance for taxonomic subsets of the same size. The noise values (depicted in Fig 2A as ribbon intervals) were applied as an additional criterion for the selection of candidate true SMAG-V9 OTU matches, as follows: the correlation or proportionality value of the candidate match should be > 0.05 higher than that of the best match from the randomly shuffled subset plus one standard deviation S1 Text in S1 File.

After examining the distribution of correlation and proportionality values for all of the V9 OTU matches for a given SMAG in our control datasets, we analysed the converse case: the set of SMAG matches for a given V9 OTU S1 Text, S3 Fig in S1 File, Fig 2B In principle, the larger the difference between ranks and proportionalities of the V9 OTU-SMAG match under consideration compared to the other SMAGs paired with the same V9 OTU, the higher the probability of the considered match being a true positive, and the others being spurious (frequently, these are artefacts of the subset size of the comparison). To obtain an estimate of the ranks and correspondence values of the false positive match pairs which we would expect to obtain for a given size of taxonomic subset, we examined the false positive matches from the simulation controls. We could then use these estimates to compare against the distribution of ranks and correspondence values of all the SMAGs matched to a given V9 OTU in the real dataset S4, S9 and S10 Figs in S1 File. Finally, we plotted the relative abundances of individual V9 OTUs and SMAGs in each candidate match to permit additional visual verification Fig 2C.

## Identification and classification of candidate matches

We calculated correlation and *rho* proportionality for each V9 OTU-MAG pair in the two lowest-rank V9 OTU taxonomic subsets for each SMAG "genre" (genre is a term used in [15] to

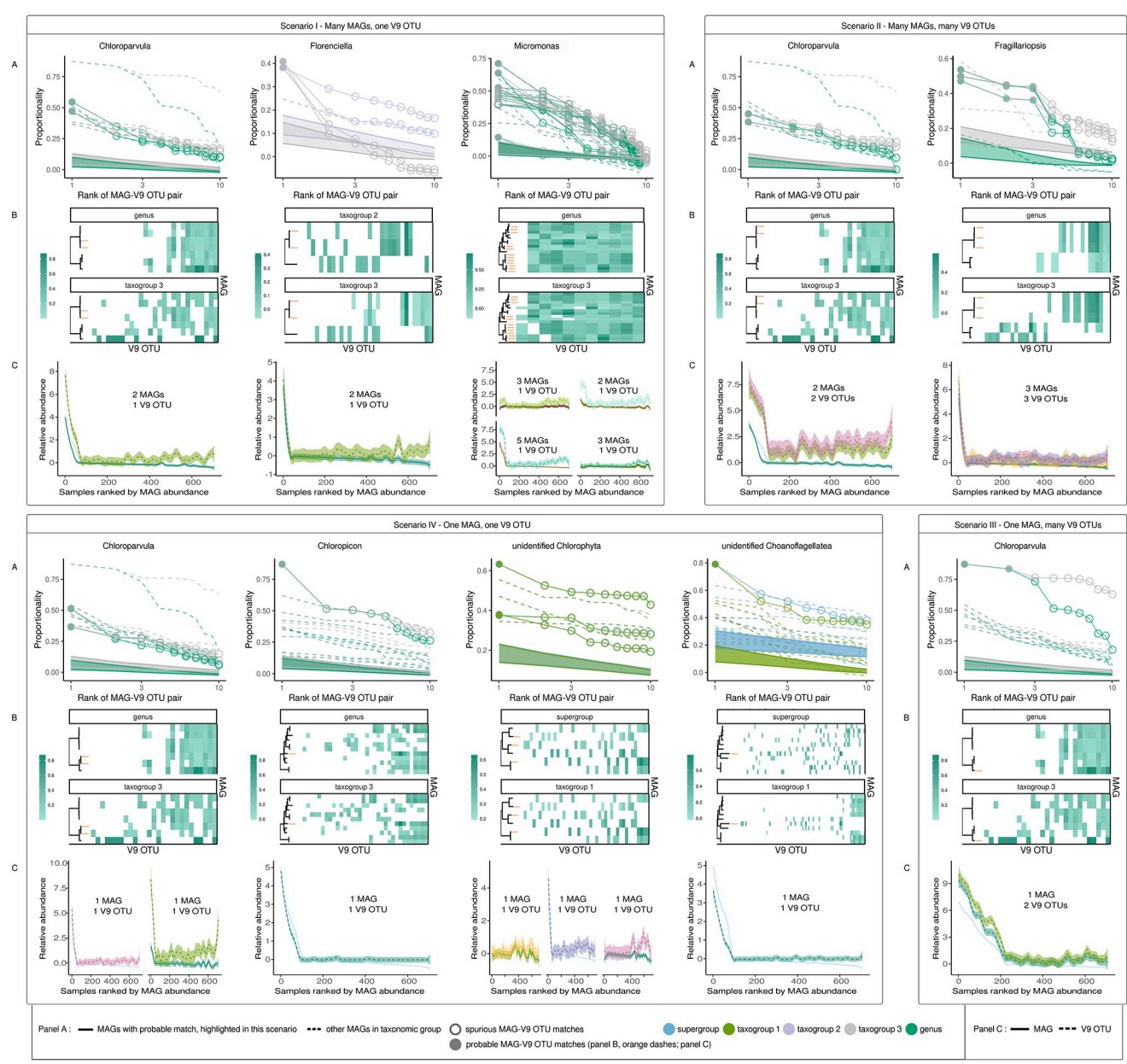

**Fig 2. Case examples illustrating different scenarios for MAG-V9 OTU match scenarios, results for the "rho" proportionality metrics (see the detailed description of each scenario in the text).** Results for Spearman's correlation are shown in S10 Fig in S1 File. A: The proportionality of the 10 best V9 OTU matches within two taxonomic subsets, for each MAG. Proportionality values of V9 OTUs paired with MAGs are shown on the vertical axis, and the rank of the match on the horizontal axis (log scale). Colours represent the rank of the V9 OTU taxonomic subset (from higher to lower rank: supergroup, taxogroup 1, taxogroup 2, taxogroup 3, or genus). Different groups have different taxonomic levels represented, depending on the resolution of the taxonomic assignments of the V9 OTUs in the group (see S1 Table in S1 File). The filled ribbons represent the mean +/- standard deviation of proportionality obtained for the shuffled datasets of a given taxonomic subset (negative controls). Lines for the MAGs highlighted in a given scenario are indicated with large points (two lines for each highlighted MAG: one for each taxonomic subset). B: The number of MAGs potentially matching a given V9 OTU. The top 10 best-ranked MAG-V9 OTU matches within the two lowest-ranked taxonomic subsets are represented as tiles. The horizontal axis represents V9 OTUs included in the taxonomic subset, the vertical axis shows MAGs, and each individual tile thus corresponds to a given MAG-V9 OTU pair. The fill colour of each tile shows the proportionality value of the match (the scale may differ among subfigures). This plot illustrates the uniqueness of each match: if several MAGs match the same V9 OTU, several tiles will be present at the same coordinate on the horizontal axis, across different points on the vertical axis and vice versa. The rank of the V9 OTU taxonomic subset presented is indicated at the top of each plot. MAGs in candidate matches from panel A are highlighted with orange dashes. Phylogenetic trees on the left of each plot represent the genetic relatedness of MAGs in each taxonomic subset, with tree topology and branch lengths from the RNA-polymerase multigene phylogeny of [15]. C: Relative abundances across stations for MAGs and V9 OTUs from candidate pairs. Each line represents a smoothed relative abundance of a single feature (either a MAG or a V9 OTU) from candidate matches. Samples on each plot are ranked based on the abundance of the MAG. Different colours represent different MAGs and V9 OTUs.

refer to the most precise taxonomic assignment of SMAGs based on manually combining evidence of taxonomic signal from an RNA polymerase phylogeny, average nucleotide identity (ANI), BUSCO gene markers and functional clustering of SMAGs [15], S1 Text, S1 Fig in S1 File; this taxonomic level frequently corresponds to genus). Next, we identified the most probable candidate matches between SMAGs and V9 OTUs based on the following 4 criteria: 1) The difference between the best match from the randomly shuffled subset negative control (its mean proportionality/correlation estimate plus its standard deviation) and the proportionality/correlation estimate of the candidate match is $> 0.05$; 2) There exists a breakpoint in proportionality/correlation between the candidate match and the next best-ranked match; 3) The candidate match appears in both of the two lowest-rank taxonomic subsets and in both correlation and *rho* proportionality metrics; 4) The candidate match is not likely to be an artefact of taxonomic subset size (i.e., subsets with only a single V9 OTU) or low occurrence/abundance of a given SMAG making it difficult for the correspondence method to distinguish (i.e., there are very few data points or very low relative abundance counts).

To mitigate for potential lack of resolution in taxonomic assignment, we equally considered the correspondences for two taxonomic subsets: both at the lowest V9 OTU rank matching the SMAG taxonomic assignment and at the next higher rank. In addition, we equally considered both correlation and *rho* estimates, and finally, we assessed the agreement between the results for different metrics and different taxonomic subsets S1 Text in S1 File.

## Results and discussion

To develop our method to detect correspondence between abundances of MAGs and metabarcodes, we made use of eukaryotic genomes and metagenomes [15] and V9 metabarcoding data [14] produced by the *Tara* Oceans expedition. We used abundance data from metagenome sequence reads mapped to both Metagenome-Assembled Genomes (MAGs) and to Single-cell Amplified Genomes (SAGs), hence, we will use the term SMAGs to refer to both together (following Delmont et al., 2022). (We used the SAGs, which may contain 18S sequences, as a positive control; see Methods) Together, the datasets we analysed contain V9 OTU [14, 48] and SMAG [15] abundances across 705 separate samples for which both the V9 region and metagenomes were sequenced.

We investigated several different methods for estimating correspondence between pairs of MAG-V9 OTU relative abundances and developed a set of controls that allowed us to validate their performance. Next, we identified candidate matching MAG-V9 OTU pairs using the selected correspondence metrics (*rho* proportionality) and Spearman's correlation and applied a set of quality criteria to distinguish the MAG-V9 OTU pairs that likely represent true matches from false-positive MAG-V9 OTU pairs (Materials and Methods).

In many cases that satisfied all quality criteria to be considered true positive matches, the correspondence between MAG and V9 OTUs within taxonomic subsets was more complex than a simple 1-to-1 relationship. Thus, we partitioned the types of relationships between MAG and V9 OTU datasets into 4 categories, depending on the number of V9 OTUs matching the same MAG, and the number of MAGs matching the same V9 OTU (illustrative example cases shown in Fig 2; all matches shown in Supplementary Information).

### Scenario I. Many MAGs match the same V9 OTU

**Criteria.** The first breakpoint appears between the top-1st and top-2nd best V9 OTU matches to a MAG, so a single top-1st rank V9 OTU match can be interpreted as the best candidate match to the MAG (in that it differs from the next best matches by having $> 0.05$ higher correlation/proportionality, as defined above). This criterion is fulfilled for more than one

MAG to the same V9 OTU. Thus, there is more than 1 MAG with the same V9 OTU as the best candidate match.

**Potential artefacts.** Scenario I can be an artefact of a low ratio of V9 OTUs to MAGs in a given taxonomic subset. The probability of true positive matches increases with the number of V9 OTUs in the taxonomic subset.

**Biological interpretation.** One biological interpretation of Scenario I could be that the V9 region does not provide enough resolution to distinguish among closely related biological entities (each of which is represented by a MAG): multiple MAGs matching the same V9 OTU actually represent the same biological entity, all sharing a similar abundance pattern that likely reflects shared ecology.

In this case, several different MAG assemblies would only differ by low-level genomic variation, which is likely not to be functional, but instead neutral (or at least not reflected in relative abundance patterns). The examples presented in Fig 2 could be interpreted as the latter case, as the genetic distance among MAGs matching the same OTU is low (see dendrograms in Fig 2B), and the relative abundance patterns are very similar Fig 2C.

Scatterplots of V9 OTU versus MAG relative abundances and genetic distances among MAGs could be examined in further detail to determine whether the variation between MAGs could be reflected in their ecological distributions.

## Scenario II. Many MAGs match many V9 OTUs

**Criteria.** The first breakpoint appears after the top-2nd (or top-3rd, or higher) match, hence, there is no possibility to discriminate the top-1st best match from the top-2nd (or higher) best match based on the difference in their proportionality/correlation estimates (Fig 2). In the comparison of whether multiple V9 OTUs are best matched to different MAGs (Fig 2B), the same set of V9 OTUs matches multiple MAGs in both taxonomic subsets. Thus, we conclude that more than one V9 OTU matches more than one MAG.

**Potential artefacts.** Scenario II is more likely to be an artefact of the taxonomic subset size if the number of both MAGs and V9 OTUs in a subset is high. Scenario II can be an artefact of extremely low abundance of MAGs or V9 OTUs (although low abundances are more common for MAGs, as metagenomic data is farther from saturation than metabarcoding data [49]). Low abundance counts do not provide enough information to distinguish among candidate matches but are nonetheless sufficient to produce proportionality/correlation values exceeding those of negative controls. Cases of false correspondence due to low abundances can be identified in pairwise scatterplots of the abundance of V9 OTUs and their MAG matches. For high-rank taxonomic subsets, Scenario II can be an artefact of the co-abundance of several unrelated biological entities, which nonetheless fall within one high-rank taxonomic group. Thus, the probability of true matches increases with increasing taxonomic resolution of the V9 OTU taxonomic subset.

**Biological interpretation.** Many V9 OTUs corresponding to many MAGs could mean that they both correspond to the same underlying biological entity, which contains high levels of variability in both the 18S V9 region and the rest of its genome (for example, a single species with genomic polymorphism, including within the ribosomal locus). In the Scenario II examples of *Chloroparvula* and *Fragillariopsis* shown in Fig 2B, the genetic distance between MAGs is low, which further supports the possibility that the MAGs may represent different haplotypes or variants within the same biological entity. In the case of *Fragillariopsis* in Fig 2B, it is particularly clear that the MAG-V9 OTU matches are highly unlikely to be false positives. However, for *Chloroparvula* the opposite is more likely: some of the V9 OTUs matching MAGs under consideration also match other MAGs within the genre with high

proportionality. Regardless, MAG-V9 OTU pairs falling within Scenario II should be considered with caution; the most likely implication is that both the V9 OTUs and MAGs involved fail to depict true underlying biological entities.

## Scenario III. One MAG matches many V9 OTUs

**Criteria.** As in Scenario II, the first breakpoint appears after the top-2nd, top-3rd (or more) match. Thus, two or more V9 OTUs are recovered as relatively equally probable matches for one MAG. In contrast to Scenario II, they are recovered uniquely for only one MAG in a genre.

We note that when MAGs are the only representatives in their genre, there is no way to discriminate Scenario II (many MAGs match many OTUs) from Scenario III.

**Potential artefacts.** Scenario III is likely to be an artefact if the ratio of MAGs to V9 OTUs in the taxonomic subset is low. All the other possible artefact sources listed for Scenario II are also true for Scenario III.

**Biological interpretation.** In biological terms, this scenario could imply that the V9 region of the SSU gene accumulates mutations faster than the corresponding genome and/or there are multiple divergent copies of the SSU gene, meaning that the diversity of V9 OTUs overestimates the actual diversity of the underlying biological entity. In this case, the V9 OTUs would be expected to have a similar relative abundance pattern across samples.

## Scenario IV. One MAG matches one V9 OTU

**Criteria.** The first breakpoint appears after the top-1st best match. The V9 OTU in the pair matches only that MAG, and no others.

MAGs which are the only representatives of their genre and have only one candidate match before the first breakpoint technically belong to Scenario IV, but as in Scenario III, their uniqueness can not be assessed appropriately, as data on other MAGs in the genre is not available. Thus, our approach cannot discriminate Scenario I from Scenario IV for subsets with only one MAG.

**Potential artefacts.** Scenario IV can be an artefact of a small number of V9 OTUs and/or MAGs in the taxonomic subset. The probability of the matches truly representing the same underlying biological entity increases with increasing the size of the taxonomic subset, for both MAGs and V9 OTUs.

**Biological interpretation.** The biological interpretation of the 1-to-1 match could be that the rate of sequence evolution of the V9 region of a biological entity is a good proxy for the rate of sequence evolution of its genome. Scenario IV is the least controversial scenario and permits the simplest and the most appealing interpretation: there is exactly one V9 OTU and one MAG in the dataset that represent the same underlying biological entity.

## Taxonomic relationship vs match scenarios

The naive assumption that each metabarcoding OTU and each metagenome-assembled genome represents a single biological entity would lead to Scenario IV (a 1-to-1 correspondence between V9 OTUs and MAGs). We observed matches not only in this scenario, but also in one-to-many, many-to-one, and many-to-many relationships; what could explain these observations?

In the case of genome assemblies, it has been shown that individual assemblies can represent ecotypes that were grouped together in the same V9 OTU [25]. However, it is also known that sequences from different divergent entities can be assembled together into one MAG [23], even for bacteria, which generally have less complex, and therefore easier to assemble,

genomes than eukaryotes [24]. It is possible to test identity by aligning the MAG to a reference genome (optimally, several reference genomes of closely related entities, to measure the variation among them) [23, 25]. However, the vast majority of protists lack reference genome sequences.

Studies based on mock communities have shown that the outcomes of metabarcoding analysis greatly depend on sampling, DNA extraction and PCR biases, factors that vary among different studies of natural samples [50], whereas other studies have shown that V9 OTUs may over- or underestimate the diversity of the underlying set of entities they represent. This can be an artefact of the clustering algorithm [26], lack of references [27] or the result of biological features of the sequenced entities. For example, V9 may not be the most suitable marker for groups including diplonemids, dinoflagellates, haptophytes, and some ciliates because it accumulates changes either too quickly or too slowly compared with rest of the genome [28, 29] or because there is copy number variation of the rRNA gene paired with sequence divergence of multiple V9 copies present in a genome [30–32]. When compared with morphological species, some studies have shown that abundance of V9 OTUs correspond with cell counts or biovolume within a given taxonomic group [11, 34, 35], whereas others produced less clear outcomes [51]. Case studies based on the amplification of V9 from a single cell have indicated that V9 OTUs most frequently contain all the amplicon diversity obtained from a single cell in radiolarians [33]. A lack of sequencing depth could also result in an underestimate of the sequence diversity present in a sample, although this is routinely tested in metabarcoding studies and is generally not a significant effect (e.g., [35]).

Here, we show that there is no uniform pattern with which MAGs match V9 OTUs. The scenarios observed are independent of the taxonomic group of the MAGs/V9 OTUs (for example, different MAGs classified within the *Chloroparvula* genus have different match scenarios) Figs 2 and 3. It is also important to compare the results across V9 OTU taxonomic subsets of different ranks. For example, a MAG in the *Fragillariopsis* genus forms a good candidate match with a V9 OTU from its corresponding "taxogroup 3" subset; however, the same V9 OTU is absent from the next-smallest taxonomic subset (genus) Fig 2B. This highlights the importance of accurate taxonomic assignments to produce optimal results within our approach.

## Outlook

Our MAG-V9 OTU matching algorithm could be applied in the future to obtain candidate MAGs matching protist laboratory cultures that have only the 18S gene sequenced, thus linking the culture to genomic information. Conversely, matching MAGs with V9 OTUs might potentially contribute an additional data source to methods for MAG taxonomic assignment. In addition, further investigation of biological explanations for different scenarios would be possible via comparison of V9 divergence versus overall genome sequence divergence, given the availability of multiple representative genomes of a low-rank taxonomic subset. However, the general lack of genome data for multiple closely related protists renders a systematic investigation of biological interpretation complicated at the present time. Finally, our algorithm for finding correspondence between the variables in two compositional (community sequencing) datasets can be tested with others of the same type.

## Conclusion

Given an appropriate set of positive and negative controls, methods to estimate correspondence, and selection criteria, some SMAGs can be matched with some V9 OTUs based on the proportionality/correlation of their relative abundances across many samples. If a single MAG

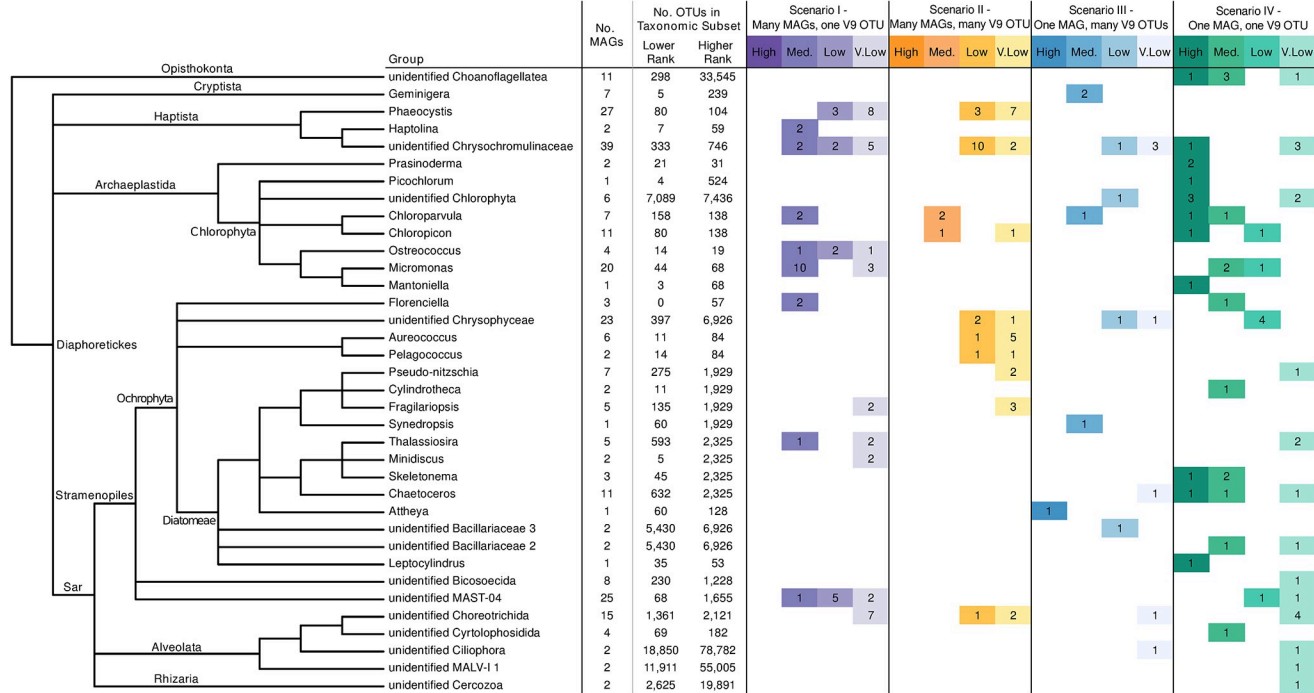

**Fig 3. Classification of MAGs into MAG/OTU match scenarios.** The taxonomic relationship (based on UniEuk; [52]) among groups of input MAGs is displayed on the left. Columns indicate: the number of MAGs in each input group, the number of OTUs in the lower-rank and second-lower-rank taxonomic subsets against which MAGs were searched for potential matches, and the distribution of MAG/OTU matches into 4 possible scenarios. Only the MAG/OTU matches meeting the criteria described in the text are included (as such, the number of classified MAGs may be lower than the total number of MAGs in a group). Within each scenario, matches are divided by confidence: high, medium, low and very low probability of being a true match.

matches a single V9 OTU, they likely represent the same biological entity. In scenarios other than a 1 to 1 match, we can enumerate hypotheses to explain the way in which multiple MAGs and/or multiple V9 OTUs may represent different underlying biological entities. If multiple V9 OTUs match a single MAG, this could mean that metabarcoding overestimates the actual diversity of the entity, and thus this scenario can result from intra-genomic polymorphism in the V9 region. If multiple MAGs match one V9 OTU, there might be intergenomic diversity within the biological entity, which is probably not reflected directly in the ecology of the entity. If a set of MAGs matches with a set of V9 OTUs, this could mean that both metagenomics and metabarcoding fail to depict true diversity and their correspondence may instead be a result of complex biological/evolutionary processes within the lineage.

## Supporting information

**S1 File. This file contains S1–S3 Texts as well as captions for S1–S10 Figs and S1–S2 Tables.** (PDF)

**S2 File. This file contains S1–S10 Figs and S1–S2 Tables.** (ZIP)

## Author Contributions

**Conceptualization:** Daryna Zavadska, Nicolas Henry, Adrià Auladell, Cédric Berney.

**Data curation:** Daryna Zavadska, Nicolas Henry, Cédric Berney.

**Formal analysis:** Daryna Zavadska.

**Investigation:** Daryna Zavadska.

**Methodology:** Daryna Zavadska, Nicolas Henry, Adrià Auladell.

**Visualization:** Daryna Zavadska.

**Writing – original draft:** Daryna Zavadska, Nicolas Henry, Adrià Auladell, Cédric Berney.

**Writing – review & editing:** Daryna Zavadska, Nicolas Henry, Adrià Auladell, Cédric Berney.

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
