## [Decision Letter · Decision Letter 0]

24 Jan 2024

PONE-D-23-39763Diverse patterns of correspondence between protist metabarcodes and protist metagenome-assembled genomesPLOS ONE

Dear Dr. Richter,

Thank you for submitting your manuscript to PLOS ONE. After careful consideration, we feel that it has merit but does not fully meet PLOS ONE’s publication criteria as it currently stands. Therefore, we invite you to submit a revised version of the manuscript that addresses the points raised during the review process.

We look forward to receiving your revised manuscript.

Kind regards,

Alberto Amato

Academic Editor

PLOS ONE

“This project has received funding from the European Research Council (ERC) under the European Union’s Horizon 2020 research and innovation programme (grant agreement No. 949745).”

“We thank Malu Calle for useful advice on statistical analyses of compositional data. This project has received funding from the European Research Council (ERC) under the European Union’s Horizon 2020 research and innovation programme (grant agreement No. 949745).”

“This project has received funding from the European Research Council (ERC) under the European Union’s Horizon 2020 research and innovation programme (grant agreement No. 949745).”

5. Please update your submission to use the PLOS LaTeX template. The template and more information on our requirements for LaTeX submissions can be found at http://journals.plos.org/plosone/s/latex.

6. Please amend the manuscript submission data (via Edit Submission) to include authors Daryna Zavadska, Adrià Auladell, Nicolas Henry, and Cédric Berney.

7. We notice that your supplementary figures and tables are included in the manuscript file. Please remove them and upload them with the file type 'Supporting Information'. Please ensure that each Supporting Information file has a legend listed in the manuscript after the references list.

Additional Editor Comments:

Dear Dr. Richter,

I apologise for the very long time we take to have your manuscript reviewed.

Both reviewers find the manuscript very well done and interesting.

Reviewers' comments:

Reviewer's Responses to Questions

**Comments to the Author**

1. Is the manuscript technically sound, and do the data support the conclusions?

Reviewer #1: Yes

Reviewer #2: Yes

2. Has the statistical analysis been performed appropriately and rigorously? 

Reviewer #1: Yes

Reviewer #2: Yes

3. Have the authors made all data underlying the findings in their manuscript fully available?

Reviewer #1: Yes

Reviewer #2: Yes

4. Is the manuscript presented in an intelligible fashion and written in standard English?

Reviewer #1: Yes

Reviewer #2: Yes

5. Review Comments to the Author

Reviewer #1: The work entitled "Diverse patterns of correspondence between protist metabarcodes and protist

metagenome-assembled genomes" by Daryna Zavadska and co-workers addresses the question of the consistency of large scale genomic studies, based on concurrent approaches, ie, either from marker gene metabarcoding (processed via the recontruction of OTUs or via ASVs) and whole genome analysis through metagenomics (including single MAG reconstructions). The authors question how these concurrent approaches compare, and if they actually lead to similar results. The comparison is made somehow difficult when some of the markers used are not in *MAG reconstructions, but using reasonable hypotheses, well selected methods, absolutely well mastered statistical analyses and real data, authors ended up with four scenarios.

Scenario I - one V9 OTU matches more than one MAG; Scenario II - more than one V9 OTU matches more than one MAG; Scenario III - more than one V9 OTU matches one MAG; Scenario IV - one V9 OTU matches one MAG. The scientific community postulating that scenario IV is the most likely, should now consider the possibility of scenarii I to III, that may leads to divergent conclusions when treating a same complex environmental sample.

This study is therefore novel, consistent, well designed, well analyzed, and extremely clearly written. I have no critics and no comment.

Reviewer #2: In the present work, Zavadska et al. try to match metagenomic assembled genomes and metabarcoding OTUs from the same TARA Oceans samples based on the correspondence of their relative abundances. They first assess which are the best proportionality/correlation metrics to use and build positive and negative controls for validation. Then, after establishing several selection criteria, they obtain 4 different types of MAG-OTU matches, which they discuss. Among other matches, the authors are able to establish reliable one-to-one correspondences in some cases, which represent valuable results to the community.

I think the study is well conducted and methods used are robust, but both methods and results could be better presented in the text and figures (more on this below).

My main concern is that in this study the V9 region (~ 100bp long) of the 18S was used, although samples for the V4 region (~ 380 bp) are also available for the TARA oceans dataset. I think that using the V4 region, which contains nearly 4 times more phylogenetic signal than V9 region, would give different results (and maybe improve the ones already obtained). Thus, I strongly suggest that the authors conduct the same analyses using the V4 dataset.

General comments:

The introduction is quite long and repetitive. A more concise writing and, in particular, avoiding repeating ideas in different paragraphs would help. Introduction and Results have several references to the methods used. Given that methods are central in this manuscript, I do not understand why is there no Methods section in the main text. It could be a good idea to create a Methods section with the main ideas instead of putting everything in the Supplementary Material. This could facilitate the reading and give a clearer structure to the manuscript. Also, it could reduce the length of other sections. For example, some lines from Introduction could go there (e.g., lines 53-55, lines 139-144), as well as some lines from the Results (a great portion of the first section, for example). Following with the structure, the Results section contains discussion parts (the discussion on each of the 4 scenarios, for example, which I think is very well presented), but then Discussion has only 3 small paragraphs. I suggest creating a single Results and Discussion section (but this is just a suggestion). Some parts from the introduction could also be moved and expanded in Discussion.

Specific comments:

· L160: where is Figure S4.4? It is constantly referenced in the main text and supplementary information but I did not find it. When clicked, it links to Figure S8. References in the text to this figure does not seem to match Figure S4 (and within Figure S4 there are no numbers but letters for each subfigure). For example, in line 368: ‘The final system used to subset SMAGs and V9 OTUs by taxonomy is summarized in Figure S4.4’.

· L177: it can be misleading to reference Figure S6 (and Figure S4.4) when Figures S2-S5 are not referenced yet (they first appear in line 181). Change Figure numbering or text order.

· L180: given that the agreement between rho and Spearman’s correlation was used to select candidate matches, these agreeing values need to be in Figure 2 instead of rho proportionality ones. If not, it can be misleading. Also, see specific section below for Figure 2.

· Figure 3: this is a really good summary of the results.

· Discussion: I miss some discussion on how the sequencing depth could affect the results presented. Are iqlr transformations enough to overcome the limitations of compositional data? Another point worth discussing is that the method used is very dependent on good taxonomic assignments. Finally, I would try to stress the results, as they are relevant to the community. Matching OTUs to MAGs seems to be difficult in most cases, but the authors were able to find a good one-to-one correspondence for some of them. This is really valuable, and I would try to emphasize it in the text.

· L311: reference?

· L350: I really appreciate that the authors are sharing all data and code. This should be the way to work in every manuscript, but unfortunately it is not.

· L380: why this filtering?

· L424: which values/thresholds were used to create these categories?

· Figure 2:

I suggest rethinking and redoing Figure 2. This figure tries to summarize so much information in a single page that it is really hard to understand. At least, it was really difficult for me to follow everything that is presented. Given that these are case examples (and not all results), it could be better to just show one representative case per scenario and save the rest for supplementary. With this, plots could be a little bit bigger and overlapping data/lines/points could be separated into different plots.

In a) all top 10 matches are shown equally and only the dots are highlighting the MAGs selected in each scenario. This is hard to interpret, as dots have too much overlap to see a clear picture. For example, the Chloroparvula plot is repeated 4 times and only the dots change (and there is a lot of overlap, so these are hard to see separately). I would make the separation between matches considered and not considered in each scenario much clearer. Having the 2 taxonomic levels mixed in the same plot is also not helpful, I would separate them.

In b), tiles are so little in some cases that are impossible to read. Also, is there some information repeated between a) and b)? This is not clear to me. Why does Florenciella have different taxonomic levels than the rest (Scenario I, section b)? This is hard to find in the text.

In c), fill colors overlap with colors of taxonomic ranks. Also, the outline of the points is hardly visible. And in some cases, the overlap is so high that the plot cannot be interpreted. Could the dots be connected to make it clearer? This needs improvement.

In d), what do the colors represent?

6. PLOS authors have the option to publish the peer review history of their article (what does this mean?). If published, this will include your full peer review and any attached files.

Reviewer #1: No

Reviewer #2: No

---

## [Author Response · Author response to Decision Letter 0]

26 Apr 2024

We responded to reviewer and editor comments in the PDF file included in this submission. That document contains appropriate formatting for the response. We include the unformatted version of our response below:

PONE-D-23-39763

Diverse patterns of correspondence between protist metabarcodes and protist metagenome-assembled genomes

Response to Reviewers

We note that all line numbers below refer to the revised manuscript document with tracked changes. This document contains LaTeX code in Microsoft Word format, which we believe is the easiest way to display the differences between the originally submitted version and this revision (despite the fact that it contains some elements of LaTeX code).

Reviewer #1: The work entitled "Diverse patterns of correspondence between protist metabarcodes and protist metagenome-assembled genomes" by Daryna Zavadska and co-workers addresses the question of the consistency of large scale genomic studies, based on concurrent approaches, ie, either from marker gene metabarcoding (processed via the recontruction of OTUs or via ASVs) and whole genome analysis through metagenomics (including single MAG reconstructions). The authors question how these concurrent approaches compare, and if they actually lead to similar results. The comparison is made somehow difficult when some of the markers used are not in *MAG reconstructions, but using reasonable hypotheses, well selected methods, absolutely well mastered statistical analyses and real data, authors ended up with four scenarios.

Scenario I - one V9 OTU matches more than one MAG; Scenario II - more than one V9 OTU matches more than one MAG; Scenario III - more than one V9 OTU matches one MAG; Scenario IV - one V9 OTU matches one MAG. The scientific community postulating that scenario IV is the most likely, should now consider the possibility of scenarii I to III, that may leads to divergent conclusions when treating a same complex environmental sample.

This study is therefore novel, consistent, well designed, well analyzed, and extremely clearly written. I have no critics and no comment.

We appreciate the reviewer’s comments and we thank them for their time in reviewing our manuscript.

Reviewer #2: In the present work, Zavadska et al. try to match metagenomic assembled genomes and metabarcoding OTUs from the same TARA Oceans samples based on the correspondence of their relative abundances. They first assess which are the best proportionality/correlation metrics to use and build positive and negative controls for validation. Then, after establishing several selection criteria, they obtain 4 different types of MAG-OTU matches, which they discuss. Among other matches, the authors are able to establish reliable one-to-one correspondences in some cases, which represent valuable results to the community.

I think the study is well conducted and methods used are robust, but both methods and results could be better presented in the text and figures (more on this below).

We thank the reviewer for their comments, which helped us to improve our manuscript. We believe we have addressed their concerns related to the methods and results, as described in our responses to each specific point below.

My main concern is that in this study the V9 region (~ 100bp long) of the 18S was used, although samples for the V4 region (~ 380 bp) are also available for the TARA oceans dataset. I think that using the V4 region, which contains nearly 4 times more phylogenetic signal than V9 region, would give different results (and maybe improve the ones already obtained). Thus, I strongly suggest that the authors conduct the same analyses using the V4 dataset.

We thank the reviewer for this suggestion. We ourselves also considered the possibility of using the V4 dataset when we initially planned our analyses. Although the V4 region is longer than the V9, we selected the V9 region for our analyses for the following 3 reasons:

The V4 region is more subject to amplification bias than is the V9 (because the V4 has significantly larger variation in length), which could potentially distort the relative abundance values we use as input.

The V4 primers used by Tara Oceans are less universal than the V9 primers used. The V4 primers are known to exclude some groups of eukaryotes and may exclude other, as yet unknown, groups.

At the time we began our analyses, the V4 data had not yet been published. We did not feel it would be appropriate for us to publish a large-scale analysis of unpublished data.

We understand the reviewer’s suggestion that V4 could provide different results or improve the ones already obtained, but we believe that adding V4 is not necessary to support the conclusions we derived from the results in our manuscript, for the following 3 reasons:

The purpose of our study was not to perform an exhaustive comparison of MAGs versus all other data sources. It was to develop a framework for pairwise comparisons of compositional datasets and, as an example case, to compare MAGs vs. V9 OTUs. In this vein, we also did not compare to other Tara Oceans datasets that could provide relative abundance data: metatranscriptomic datasets (within which ribosomal sequences can be collected), or imaging data.

Due to the high computational burden of performing the simulations necessary to produce benchmarks for proportionality and correlation values, we estimate that adding a V4 analysis would take 3-4 months, only considering computational time.

Variable regions in the 18S do not necessarily evolve at strictly the same rate, even between closely related species within a single eukaryotic lineage. In a small minority of eukaryotic lineages, we expect a perfect one-to-one V4-V9 correspondence. However, in most cases we expect that there is either one V4 for multiple V9s or multiple V4s for one V9. Furthermore, in the case where one V4 corresponds to one V9, it may still hide many different “species” based on ITS or other more variable markers. As the reviewer alluded, this would change the nature of our analyses from a pairwise statistical comparison, to a multiple statistical comparison among 3 datasets, and would necessitate a new framework to classify and adjudicate differences between V4 and V9. While we agree that this would be a valuable contribution, it would likely require a significant change to our manuscript, and may not necessarily be useful in the near future, as many projects are already adopting long-read metabarcoding approaches that include both variable regions (e.g., Jamy et al., 2022 https://doi.org/10.1038/s41559-022-01838-4; Gaonkar and Campbell, 2024 https://doi.org/10.1002/ece3.11232).

Therefore, although we agree with the reviewer’s suggestion that adding V4 could contribute to further knowledge on the relationship between MAGs and barcoding datasets, we believe that adding V4 analyses would be out of the scope of our current work. We would instead suggest that future authors perform comparisons to V4 (or to other datasets), for which we believe our methodology could serve as a basis.

General comments:

The introduction is quite long and repetitive. A more concise writing and, in particular, avoiding repeating ideas in different paragraphs would help. Introduction and Results have several references to the methods used. Given that methods are central in this manuscript, I do not understand why is there no Methods section in the main text. It could be a good idea to create a Methods section with the main ideas instead of putting everything in the Supplementary Material. This could facilitate the reading and give a clearer structure to the manuscript. Also, it could reduce the length of other sections. For example, some lines from Introduction could go there (e.g., lines 53-55, lines 139-144), as well as some lines from the Results (a great portion of the first section, for example). Following with the structure, the Results section contains discussion parts (the discussion on each of the 4 scenarios, for example, which I think is very well presented), but then Discussion has only 3 small paragraphs. I suggest creating a single Results and Discussion section (but this is just a suggestion). Some parts from the introduction could also be moved and expanded in Discussion.

We thank the reviewer for these suggestions. To facilitate reading following the reviewer’s advice, we (1) created a new Results and Discussion section, (2) moved parts of the Introduction to the new combined Results and Discussion section, (3) created a Methods section and moved parts of the Introduction and of the Results into the new Methods section, and (4) reorganized the Introduction to remove repetitive sections.

Specific comments:

· L160: where is Figure S4.4? It is constantly referenced in the main text and supplementary information but I did not find it. When clicked, it links to Figure S8. References in the text to this figure does not seem to match Figure S4 (and within Figure S4 there are no numbers but letters for each subfigure). For example, in line 368: ‘The final system used to subset SMAGs and V9 OTUs by taxonomy is summarized in Figure S4.4’.

We thank the reviewer for pointing out this error. This was a bug in the LaTeX file that we used to produce the submitted version of the manuscript. We corrected the error in the resubmitted manuscript.

· L177: it can be misleading to reference Figure S6 (and Figure S4.4) when Figures S2-S5 are not referenced yet (they first appear in line 181). Change Figure numbering or text order.

We corrected these errors in figure numbering.

· L180: given that the agreement between rho and Spearman’s correlation was used to select candidate matches, these agreeing values need to be in Figure 2 instead of rho proportionality ones. If not, it can be misleading. Also, see specific section below for Figure 2.

We tried the reviewer’s suggestion to include Spearman’s correlation along with rho proportionality values in Figure 2, but we could not find a way to present all of the data in a clear and readable manner. Instead, we placed all of the Spearman’s correlation data in the supplementary figures. We also note that all data necessary to select candidate matches (including proportionality and correlation data) is available in a table on the GitHub accompanying our manuscript, here: https://github.com/beaplab/Protist_barcode_MAG_correspondence/blob/main/data/downstream_outputs/Candidate_pairs-for_R_script_scatterplot_corrected_Pairs_automated.csv

Please see below for responses to other specific suggestions on Figure 2.

· Figure 3: this is a really good summary of the results.

Thank you for this comment, we are glad that the figure provides a useful summary of the overall results.

· Discussion: I miss some discussion on how the sequencing depth could affect the results presented. Are iqlr transformations enough to overcome the limitations of compositional data? Another point worth discussing is that the method used is very dependent on good taxonomic assignments. Finally, I would try to stress the results, as they are relevant to the community. Matching OTUs to MAGs seems to be difficult in most cases, but the authors were able to find a good one-to-one correspondence for some of them. This is really valuable, and I would try to emphasize it in the text.

Thank you for the suggestions. We did not test the effects of sequencing depth in our study, since we used data from a previous paper that directly tested the effects of sequencing depth on the recovery of community diversity in the form of OTUs (Figure 1A and associated supplementary figures from DOI: 10.1126/science.1261605). We added a sentence to the Discussion (lines 852-855) to address the reviewer’s suggestion: “A lack of sequencing depth could also result in an underestimate of the sequence diversity present in a sample, although this is routinely tested in metabarcoding studies and is generally not a significant effect”

Regarding whether iqlr transformations are sufficient to overcome the limitations of compositional data: we agree that although iqlr transformations are theoretically intended to overcome these limitations, it is useful to know whether the approach is valid in practice. Following iqlr transformation, we produced results from both shuffling of actual data and using simulated datasets that showed that we were able to distinguish signal from noise (in which the controls represented a noise signal). Nonetheless, the signal that we were able to detect may be due to an actual biological pattern or may be the results of multiple biological biases. Our results matching V9 OTUs with SAGs (as positive controls) suggest that the biological patterns can indeed be detected for some taxonomic groups, but without a larger number of positive control SAGs, we were not able to test this exhaustively. For other taxonomic groups, we were not able to match V9 OTUs with positive control SAGs, as their proportionality or correlation values were below those of artificially generated noise; in these cases, either the underlying biological signal was not strong enough for our method (including the iqlr transformation) to detect it, or the number of stations in which the SAG was present was too low (as few as only 5 stations), or some combination of the two possibilities (see Supplementary Figure 8).

We completely agree with the reviewer’s point that good taxonomic assignments are critical for our method to work properly. To emphasize this, we added a sentence to the Discussion, lines 864-865: “This highlights the importance of accurate taxonomic assignments to produce optimal results within our approach.”

We also appreciate the reviewer’s point that our approach was able to identify potential one-to-one matches between V9 OTUs and MAGs, and that could be valuable to the community. The philosophy guiding how we prepared our manuscript was to present the results, with appropriate caveats, and to allow the reader to draw their own conclusions based on these results. Therefore, we tried to refrain from overemphasizing the potential V9 OTU and MAG matches that we found in the text, as we believe they represent initial hypotheses for potential relationships. We would not want to lead the reader to believe that we presented strong, direct evidence for any individual match, nor that the overall approach is generally applicable to all taxonomic groups (since we hope to encourage researchers working on specific taxonomic groups to investigate the results of our approach within that particular group). 

· L311: reference?

Although the idea that either V9 OTUs or MAGs are more appropriate representations of underlying biological entities is implied in various articles in the literature, we were unable to find any references that explicitly assert the superiority of one or the other approach (instead, it is generally implicit, in our interpretation). To reduce the possibility of confusion, we removed this sentence from our article.

· L350: I really appreciate that the authors are sharing all data and code. This should be the way to work in every manuscript, but unfortunately it is not.

We completely agree.

· L380: why this filtering?

We removed V9 OTUs with total abundance less than 100 as they were not likely to contribute any signal to our measurements of proportionality or correlation. This is reflected in the figure below, where total abundance for reads assigned to a given OTU is on the horizontal axis and that OTU’s maximum relative abundance in a single sample is on the vertical axis. As can be observed, OTUs with total abundance less than 100 essentially never contribute substantially to relative abundance in any sample:

To address the reviewer’s comment, we added the following to the supplementary information (lines 1339-1342): “we removed V9 OTUs whose sum of number of reads across all samples was ≤ 100 from subsequent calculations, as these OTUs did not contribute substantially to relative abundance in any sample (data not shown).” We note that we corrected a typo in this sentence, in which the threshold was erroneously written to be ≥ 100, not ≤ 100. We thank the reviewer for pointing out this error.

· L424: which values/thresholds were use

---

## [Editor Report · Decision Letter 1]

30 Apr 2024

Diverse patterns of correspondence between protist metabarcodes and protist metagenome-assembled genomes

PONE-D-23-39763R1

Dear Dr. Richter,

We’re pleased to inform you that your manuscript has been judged scientifically suitable for publication and will be formally accepted for publication once it meets all outstanding technical requirements.

Kind regards,

Alberto Amato

Academic Editor

PLOS ONE

Additional Editor Comments (optional):

Dear Dr; Richter,

thank you very much for the great job you and your colleagues did on the manuscript.

I am glad to accept your revised version for publication.

Sincerely

Alberto Amato
---

## [Editor Report · Acceptance letter]

10 May 2024

PONE-D-23-39763R1 

PLOS ONE

Dear Dr. Richter, 

I'm pleased to inform you that your manuscript has been deemed suitable for publication in PLOS ONE. Congratulations! Your manuscript is now being handed over to our production team.

Kind regards, 

on behalf of

Dr. Alberto Amato 

Academic Editor

PLOS ONE